# Oligoclonal T Cells Transiently Expand and Express Tim-3 and PD-1 Following Anti-CD19 CAR T Cell Therapy: A Case Report

**DOI:** 10.3390/ijms19124118

**Published:** 2018-12-19

**Authors:** Christopher Ronald Funk, Christopher T. Petersen, Neera Jagirdar, Sruthi Ravindranathan, David L. Jaye, Christopher R. Flowers, Amelia Langston, Edmund K. Waller

**Affiliations:** 1Department of Hematology and Oncology, Emory University School of Medicine, Atlanta, GA 30322, USA; ronnie.funk@emory.edu (C.R.F.); chris.petersen@stjude.org (C.T.P.); njagirdar@cambiumbio.com (N.J.); sruthi.ravindranathan@emory.edu (S.R.); crflowe@emory.edu (C.R.F.); alangst@emory.edu (A.L.); 2Department of Pathology and Laboratory Medicine, Emory University School of Medicine, Atlanta, GA 30322, USA; dljaye@emory.edu

**Keywords:** CTL019, tisagenlecleucel, oligoclonal T cell expansion, DLBCL, T cell immunoglobulin mucin domain 3 (Tim-3), programmed cell death protein 1 (PD-1)

## Abstract

Clinical trials of chimeric antigen receptor (CAR) T cells in hematologic malignancy associate remissions with two profiles of CAR T cell proliferation kinetics, which differ based upon costimulatory domain. Additional T cell intrinsic factors that influence or predict clinical response remain unclear. To address this gap, we report the case of a 68-year-old woman with refractory/relapsed diffuse large B cell lymphoma (DLBCL), treated with tisagenlecleucel (anti-CD19), with a CD137 costimulatory domain (4-1BB) on an investigational new drug application (#16944). For two months post-infusion, the patient experienced dramatic regression of subcutaneous nodules of DLBCL. Unfortunately, her CAR T exhibited kinetics unassociated with remission, and she died of DLBCL-related sequelae. Serial phenotypic analysis of peripheral blood alongside sequencing of the β-peptide variable region of the T cell receptor (TCRβ) revealed distinct waves of oligoclonal T cell expansion with dynamic expression of immune checkpoint molecules. One week prior to CAR T cell contraction, T cell immunoglobulin mucin domain 3 (Tim-3) and programmed cell death protein 1 (PD-1) exhibited peak expressions on both the CD8 T cell (Tim-3 ≈ 50%; PD-1 ≈ 17%) and CAR T cell subsets (Tim-3 ≈ 78%; PD-1 ≈ 40%). These correlative observations draw attention to Tim-3 and PD-1 signaling pathways in context of CAR T cell exhaustion.

## 1. Introduction

Pharmacokinetic and pharmacodynamic properties of drugs play a salient clinical role by guiding strategies to achieve therapeutic dosage. Given that chimeric antigen receptor (CAR) T cells represent a new class of cellular therapy, numerous facets of CAR T cell kinetics and dynamics, which may influence clinical response, remain unclear. The majority of clinical trials associate durable persistence of CAR T cells with remission, particularly for constructs containing a CD137 (4-1BB) co-stimulatory domain [1,2]. High expression of CAR by quantitative polymerase chain reaction (PCR) was observed two years following an initial remission, to suggest long-term CAR T persistence is necessary to produce durable remissions [2]. Consistent with this theory, the long-term sequela of CD19 cell aplasia, observed across numerous clinical trials, likely reflects persistent on-target effector function [2]. In contrast to these studies, a single-center phase I clinical trial of 53 patients with B-cell acute lymphoblastic leukemia (B-cell ALL) using CD28 co-stimulatory domain, observed that durable persistence of CAR T did not correlate with short- or long-term remission. Instead, the ratio of CAR T cell peak expansion to tumor burden positively correlated with overall survival and remission [3]. The differing CAR T cell expansion and contraction kinetics suggests that additional characterization of intrinsic T cell qualities and dynamics during CAR T cell therapy may yield opportunities to optimize clinical outcomes.

A current challenge limiting the ability of CAR T cells to induce remissions for all patients with hematologic malignancy is characterization of CAR T cell kinetics in patients who both respond and fail to respond to therapy. In this study, we characterize the expansion of T cells from a patient with diffuse large B cell lymphoma (DLBCL) who received an anti-CD19 CAR T construct containing a 4-1BB co-stimulatory domain (CTL019) [4]. We herein report transient expansion of distinct waves of oligoclonal T cells, with dynamic increases in expression of immune checkpoint inhibitors Tim-3, PD-1, and lymphocyte activation gene 3 protein (LAG3) prior to the contraction phase of the T cell response.

## 2. Case Description

A 68-year-old woman with chronic kidney disease stage II presented with worsening sacral pain in 2012. Evaluation revealed multiple lumbosacral foci of DLBCL. Disease persisted despite induction chemotherapy with rituximab, cyclophosphamide, doxorubicin hydrochloride, vincristine sulfate, and prednisone alongside lumbosacral radiation. Salvage chemotherapy with rituximab, ifosfamide, carboplatin, and etoposide, supplemented with lumbosacral radiation, achieved a PET-negative complete remission.

Months later, the patient noticed a subcutaneous nodule superficial to her right scapula, and biopsy showed recurrent DLBCL. After surgical resection and adjuvant gemcitabine, rituximab, and oxaliplatin, her DLBCL remained refractory to therapy. She was enrolled in a phase II trial (clinicaltrials.gov #NCT02445248) assessing CTL019 in DLBCL (JULIET) [4]. Leukapheresis and CAR T manufacture were successful, but she developed postmenopausal vaginal bleeding, heralding diagnosis of stage I endometrial carcinoma which precluded further participation in JULIET. A compassionate-use IND application (#16944) was approved given CTL019 manufacture occurred prior to symptoms of endometrial carcinoma. CTL019 was infused following three days of lymphocyte-depleting fludarabine and cyclophosphamide. At this time, six subcutaneous nodules were present dorsal to her right scapula DLBCL, clinically consistent with recurrent DLBCL. She tolerated the CAR T infusion well, with no side effects, and was discharged three days later.

Her post-CAR infusion course was complicated by three presentations of neutropenic fever with autonomic instability and pancytopenic aplasia. She lacked described [5] neurologic or general symptoms concerning cytokine release syndrome (CRS), neither did she develop any signs of end organ failure associated with CRS. Laboratory evaluation showed nonspecific signs of inflammation: ferritin, 864–1946 ng/mL (normal 11–307 ng/mL); lactose dehydrogenase, 98–215 units/L (normal <200 U/L); and interleukin-6 (IL-6), 12–19 pg/mL (normal <5 pg/mL). Her neutropenic fevers were each considered consistent with septic shock given positive blood and urinary cultures for *Enterobacter cloacae* treated with ciprofloxacin. Observation of the subcutaneous deposits of DLBCL showed regression of palpable lesions over the two months following CAR T infusion, with local breakdown of the skin over one of the lesions (Figure 1).

Peripheral blood was collected for analysis on post-infusion days 1, 8, 17, 21, 31, and 58. T cell populations peaked by day 31 (Figure 2A–D). CAR T cells accounted for 0.4% of the total CD3 expressing cell population at day 17. T cell immunoglobulin mucin domain 3 (Tim-3), was expressed on more cells than programmed cell death protein 1 (PD-1), with peak expressions on both the CD8 T cell (Tim-3 ≈ 50%; PD-1 ≈ 17%, Figure 2G) and CAR T cell subsets (Tim-3 ≈ 78%; PD-1 ≈ 40%, Figure 1H). Tim-3 was preferentially expressed on the CD8 subset, while lymphocyte activation gene 3 protein (LAG3) was more expressed on the CD4 subset, although on <10% of clones (Figure 2F). Immune checkpoint inhibitor overexpression was greatest on day 8, concurrent to CAR T cell expansion, but preceding a T cell contraction phase from day 20 onward (Figure 2E–H).

In order to determine the effects of CAR T expansion on other immune cells in the blood, the frequencies and phenotypes of other immune cells, at the peak of T cell expansion on day 31 post CAR T, were characterized by flow cytometry, as shown in Figure 2. These data show that even at the time of peak T cell expansion, numbers of CD3+ T cells remained low (Figure 3A). CD4+ T cells comprised 10.8% of the mononuclear cell population and 29.3% of all mononuclear cells were CD3+ CD8+ (Figure 3B). After infusion of anti-CD19 directed CAR T, little to no CD19 expressing cells were detected, suggesting on-target CAR T function (Figure 3C). The increase in CD56^bright^ CD16-cells (Figure 3D) likely represents an increase in cytolytic NK (natural killer) cells, whereas the increase in CD56^dim^ CD16+ cells represent NK cells with replicative potential, as reviewed [6]. CD56^bright^ CD16+ cells are thought to represent a population of cytotoxic T cells, with both αβ and γδ T cells expressing these antigens [6]. Populations of macrophages and immature monocytes (CD14^dim^ expression, Figure 3E) were increased following CAR T administration. In summary, these data in combination with a dramatic regression of subcutaneous nodules of DLBCL, apparent on examination, and confirmed by PET/CT, suggested on-target CTL019 function in depleting CD19+ targets.

To evaluate her prolonged pancytopenia (detected day 31 post-CAR T), which required repeated platelet and blood transfusions, a bone marrow aspirate was performed and immunophenotyping of marrow cells was compared to peripheral blood in Figure 4. The total cellular content of bone marrow was significantly reduced across all lymphocytes, including CD3 positive cells (Figure 4A). Anti-CD19 CAR T cells within both the CD4 and CD8 subsets remained detectable in the peripheral blood (Figure 4B,C), and these CAR T were scarce in the marrow. Lastly, ratios of naïve (CD45RA+) and memory/activated T cells (CD45RO+) were observed to be nearly identical in both the peripheral blood and bone marrow (Figure 4D); however, the total quantity of cells in bone marrow was reduced. Of note, during pancytopenic aplasia, the ratio of CD8+ CD27-/CD28-cells increased (Figure 4D).

To assess the clonality of the global T cell compartment, deep sequencing of the T cell receptor-β (TCRβ) complementarity determining region-3 (CDR3) was performed (Figure 5A). Clones of T cells with a productive frequency of 5% or less prior to CAR T administration were identified and increases in productive-frequency were tracked post-infusion, with Figure 5A showing clones with the largest increases. A concomitant increase in CAR T was observed using flow cytometry (Figure 5, Vβ-20). Figure 5B shows T cell clones collected from the DLBCL nodules on day 10. Clones present at higher frequency in the tumor, present to a lesser extent in blood, suggest anti-tumor specificity of these T cell clones (purple dots, Figure 5B) common to both compartments. Small numbers of clones expanded at high frequencies and homed to the tumor nodule, which are denoted by purple boxes (Figure 5B). Despite these findings, CTL019 therapy failed to induce a complete and durable response for this patient. She later received monoclonal antibody against PD1, which also failed to induce a remission. She then opted for supportive care, and died from sequelae of DLBCL six months later.

## 3. Discussion

Clinical assessments of this patient and analysis of B-cells in the blood indicated an on-target response to therapy. CD19 expression on B lymphocytes was detectable following lymphodepleting chemotherapy, prior to administration of CAR T, but became undetectable by day 31 post-infusion (Figure 3C). While the patient exhibited a partial response to CTL019; tisagenlecleucel therapy did not lead to a durable remission, raising important questions regarding the characteristics of CAR T expansion or phenotypes that may be associated with failure to eradicate the DLBCL.

There are a number of limitations to this study. First, it represents the analysis of only a single patient, and results from this one patient may not be generalizable to larger numbers of patients treated with CAR T. Nevertheless, results presented here should be viewed as hypothesis generating that could be prospectively evaluated in larger numbers of patients undergoing CAR T therapy. Second, while we characterize the CAR T in the blood phenotypically, we could not directly correlate the phenotypical analysis of CAR T with deep sequencing of the TCR and identification of oligoclonal expansion of T cells post CAR T infusion. However, the marked expansion of T cell clones up to frequencies of 10% of blood T cells immediately following CAR T infusion (Figure 5A), suggests that at least some of these T cell clones represented expansion of CAR T. Third, detailed clinical studies of this patient were limited to the first two months post CAR T. Nevertheless, the major clinical response to therapy and expansion and contraction of CAR T and T cell clones defined by TCR sequencing occurred during this same time frame, making the observations presented herein of clinical interest.

At the peak of CAR T expression (day 17), the CAR T cells in this patient comprised approximately 0.43% of the total CD3 population (calculated from Figure 2), with a peak number of CAR T/µL of less than 2 cells/µL. This low frequency and absolute numbers of CAR T in this patient suggests an insufficient ratio of CAR T to tumor cells to achieve complete tumor eradication, given that the three non-responders to CAR T therapy in a phase I trial in B-ALL had CAR T cells that comprised only 0.2%, 0.6%, and 8.2% of the total CD3 cells, whereas CAR T comprised 39.8% of total CD3 cells for complete responders [2]. A log-normal decay profile has been described as a pattern associated with remission for 4-1BB costimulatory domain CAR T cells in chronic lymphocytic leukemia (CLL) [7]. This patient’s CAR T cells peaked in the blood later than has been described for tisagenlecleucel therapy in DLBCL and at lower peak levels [8], and appeared to decay with linear slope, by flow cytometry based upon multiple staining methodologies (Figure 1D and Figure 5). The absence of a more sensitive PCR test for the presence of CAR T and the lack of follow-up later than 60 days post-infusion represents a limitation in defining the long-term persistence of CAR T in this patient. The results in this patient with DLBCL support that a threshold of CAR T as a percentage of total CD3 may represent a pharmacodynamic property to monitor in future clinical trials.

A second factor related to CAR T failure is senescence of the T cells prior to and during CAR T manufacturing and post-infusion. The observed increase in a senescent population of CD27-, CD28-cells in the bone marrow during cytopenic aplasia in this patient (Figure 4D) is notable given that loss of co-expression of these markers may represent an immunosuppressive state of T cell senescence [8,9,10]. We have previously shown that removal of CD27-/CD28-T cells prior to ex vivo T cell expansion led to doubled T cell expansion at 14 days [11]. More recently, frequencies of CD27+ CD45RO- CD8+ T cells prior to CAR T manufacture were positively associated with sustained response in CLL, putatively because this population represents antigen-specific T cells that expand rapidly upon re-exposure to antigen [12]. Therefore, loss of CD27/CD28 co-expression on the majority of T cells in the marrow (a site of CD19 antigen from normal B cells) may have contributed to failure of the CAR T to expand robustly and the contraction of the entire T cell populations seen in Figure 2.

The heightened levels of co-inhibitory molecule expression on the CAR T in the blood of this patient following CAR T infusion represent a third mechanism by which CAR T may lack efficacy. Over-expression of Tim-3 on 78% of the CAR T cells provides correlative evidence that Tim-3 signaling may have contributed to linear decay of the CAR T cells in this patient. Since this patient received CTL019 on a single patient IND using CAR T manufactured in the same way as for the clinical trial, JULIET, comparison of data from this single patient to responders in the JULIET study is possible. Baseline biopsies of DLBCL nodules were analyzed by quantitative immunofluorescence and AQUA analysis, to reveal complete responders in JULIET exhibiting a median 33% expression of Tim-3, median 12% expression of LAG-3, and median 1.9% PD-1 expression on the total CD3 compartment [13]. Partial responders yielded lower baseline median expression of immune checkpoint molecules on total CD3 cells (Tim-3 ≈ 20%, LAG3 ≈ 8%, and PD-1 ≈ <1%), suggesting elevations in immune checkpoint expression may reflect enhanced T cell responses to tumor antigens at baseline. Whether these baseline data representing immune checkpoint expression on total CD3 correlate directly to CAR T cell expression profiles is unknown. For this single patient, the proportion of Tim-3 expression on CAR T cells at day 1 was double the proportion of Tim-3 expression on total CD3. This ratio of Tim-3 expression on CAR T to non-CAR T was maintained throughout the 60-day study period, suggesting that assessment of initial immune checkpoint expression on the CAR T, in combination with monitoring immune checkpoint expression on total CD3, reflects a monitoring strategy during CAR T cell therapy worthy of further exploration. Nonetheless, since Tim-3 expression on CAR was greater than on total CD3, blockade of Tim-3 and/or PD-1 signaling may mitigate the T cell exhaustion that occurs amidst repetitive exposure of CAR T cells to antigen [14], to potentially enhance overall outcomes. Furthermore, given that Tim-3 was preferentially expressed on the CD8 subset and LAG3 on the CD4 subset, inhibition of Tim-3 or LAG3 may alter the ratio of CD8:CD4 T cells in efforts to optimize outcomes.

Finally, the pattern of oligoclonal T cell expansion following CAR T infusion indicates that not all CAR transductions result in T cells that expand equally in vivo. Proliferation in response to antigen is known to potentially induce oligoclonal or clonal T cell expansion in response to malignancy, as reviewed [15]. However, in CAR T cell therapy, all cells transduced with CAR theoretically receive activating signal from antigen, rendering a polyclonal expansion more likely. Further, during CAR T manufacture, no sorting of the apheresis product was performed prior to transduction. Thus, the CAR was transduced across the gamut of polyclonal CD3 cells. In summary, given that this patient received 1 × 10^8^ CD3+ polyclonal CAR T cells, the observation that a single clone could comprise nearly 10% of the peripheral T cell productive frequency is remarkable. Potential explanations are guided by the observation that the patient’s T cell compartment exhibited a Gaussian distribution prior to CAR T administration, to suggest DLBCL-mediated immunosuppression did not explain the oligoclonal expansion. Recently, CAR T cells with a memory phenotype gene expression profile (IL-6, STAT3) were associated with remissions, thereby suggesting that CAR T cell intrinsic differences likely underlie the oligoclonal expansion observed in this patient [12]. The observations made from this patient serve to drive the formation of hypotheses which aim to further immunotherapy. The global decrease in CD27/28 expression and the over-expression of Tim-3 on 78% of CAR T cells may explain the oligoclonal expansion of T cells in this patient. Further investigation of the generalizable significance of these observations to other people receiving CAR T cell therapy is warranted.

## 4. Materials and Methods

### 4.1. Mononuclear Cell Samples

Peripheral blood and bone marrow samples were collected prior to and following infusion of CD19 CAR T cells and mononuclear cells were isolated and frozen as previously described [11].

### 4.2. Flow Cytometry

Frozen mononuclear cell samples were thawed and suspended in flow cytometry buffer at a concentration of 10^7^ cells/mL. After eliminating non-specific binding of antibodies using F_c_ block (Miltenyi, Bergisch Gladbach, Germany), the cells were stained with the following fluorescently-labeled monoclonal antibodies: PE-CF594 anti-CD3, APC-Cy7 anti-CD4, FITC anti-CD8, BV421 anti-Tim-3, PE anti-LAG-3 (BD Biosciences, San Jose, CA, USA), and PE-Cy7 anti-PD-1 (eBioscience, San Diego, CA, USA). Identification of CD19 CAR T cells was performed by first staining with a biotinylated CD19-IgG Fc fusion protein (Acrobiosystems, Newark, DE, USA) followed by staining with streptavidin-APC (eBioscience, San Diego, CA, USA).

### 4.3. TCR Deep Sequencing

DNA samples extracted from frozen mononuclear cell aliquots (Adaptive Biotechnologies, Seattle, WA, USA) were run on immunoSeq platform and amplified using a multiplexed PCR reaction with Vβ and Jβ primers. TCRβ CDR3 regions were sequenced using Illumina Genome Analyzer station and analyzed with ImmunoSeq Analyzer.

### 4.4. Ethics Approval and Consent

This study was approved by the Emory Institutional Review Board (#88462, 13 January 2015) under a compassionate use IND (#16944, 30 March 2016).

### 4.5. Consent for Publication

The patient provided written consent for de-identified reporting and written analysis of all results, which is on file in the Emory University Bone Marrow Transplant Clinical Trials Office.

## Figures and Tables

**Figure 1 ijms-19-04118-f001:**
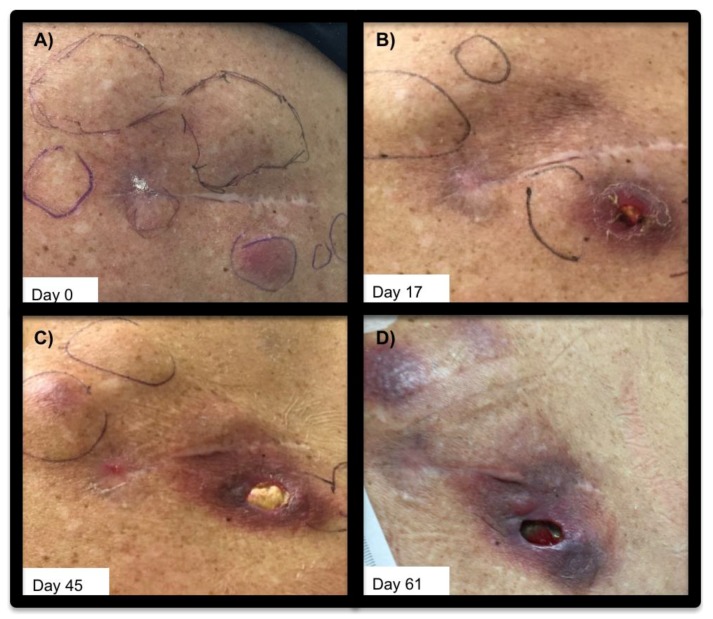
Subcutaneous DLBCL lesions pre- and post- CAR T cell infusion. Subcutaneous DLBCL lesions superficial to right scapula, shown (**A**): prior to CAR T infusion (day 0) (**B**): 17 days post-infusion of CAR T cells, (**C**): 45 days post-CAR T, and (**D**): day 61 post-CAR T infusion. Left is medial, and right is lateral.

**Figure 2 ijms-19-04118-f002:**
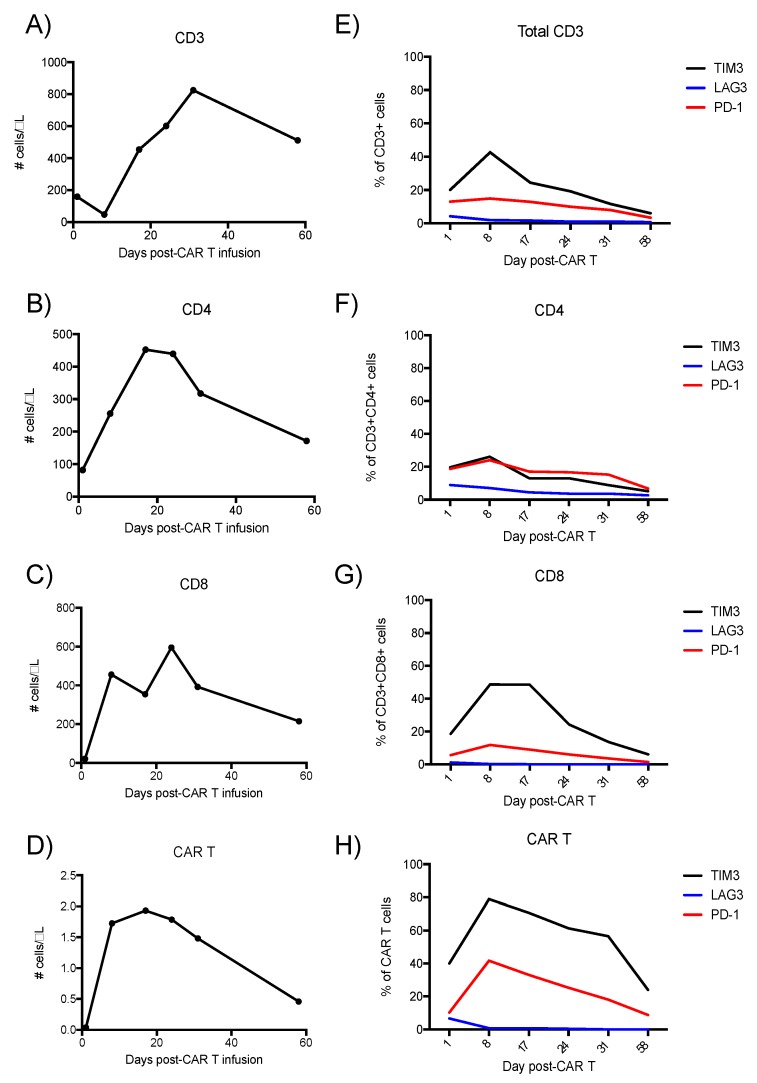
Transient expansion of T-cells and CAR T cells after tisagenlecleucel CAR T infusion. Panels A–D: Flow cytometry of PBMCs derived from peripheral blood, assessing expression of (**A**): CD3 (**B**): CD4 (**C**): CD8, and (**D**): CAR. Panels (**E**–**H**): Expression of immune checkpoint regulators on the T-cells over the same 58 days post-tisagenlecleucel infusion.

**Figure 3 ijms-19-04118-f003:**
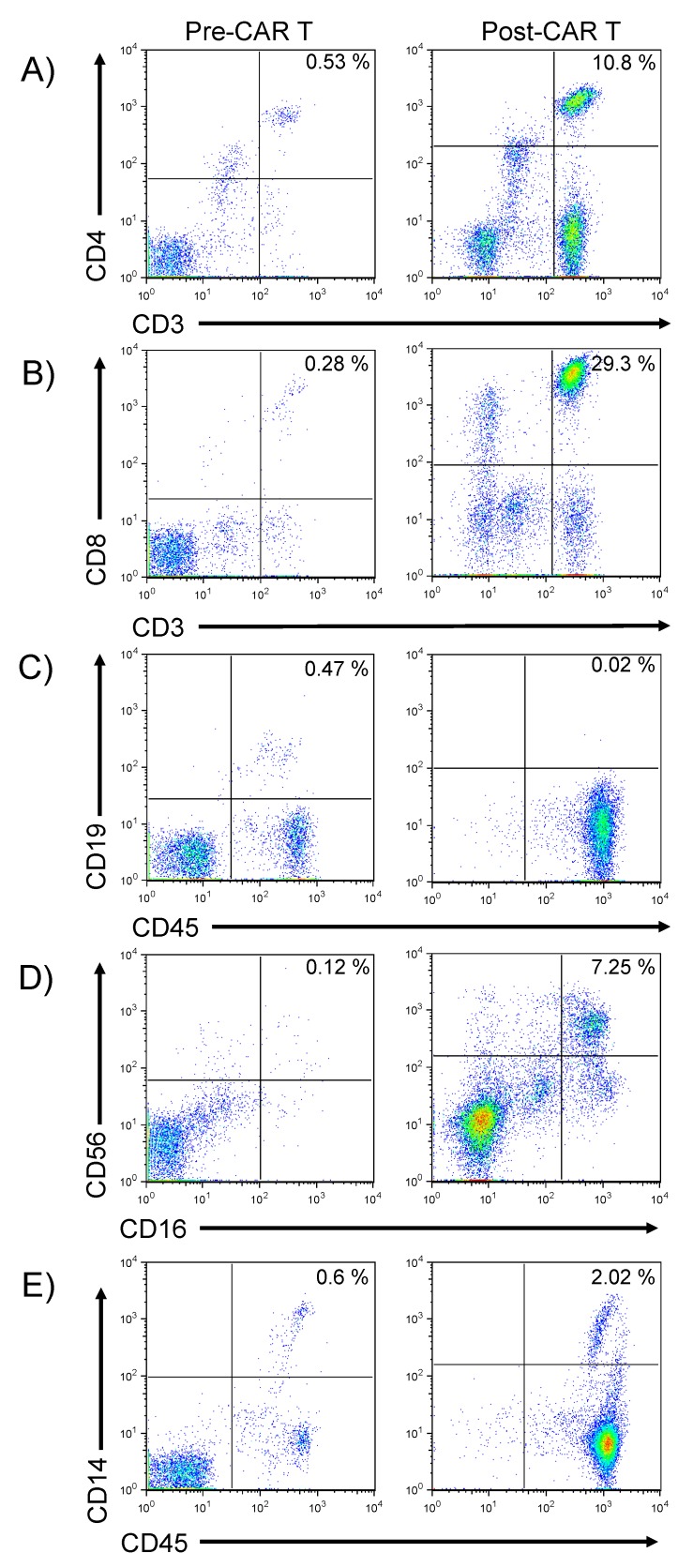
Phenotypic analysis of peripheral blood pre- and post-CAR T infusion. Flow cytometry was performed upon peripheral blood to enable phenotypic analysis of immune system cells, including (**A**): expression of CD3 and CD4 on T cells, (**B**): expression of CD3 and CD8 on T cells, (**C**): CD19 and CD45 expression on B-lymphocytes (**D**): CD56 and CD16 expression on NK cells or cytotoxic T cells, and (**E**): expression of CD14 and CD45 on monocytes. “Pre-CAR T” denotes analysis performed following lymphodepleting chemotherapy but prior to administration of CAR T cells, whereas “Post-CAR T” denotes analysis on post-CAR T cell infusion day 31.

**Figure 4 ijms-19-04118-f004:**
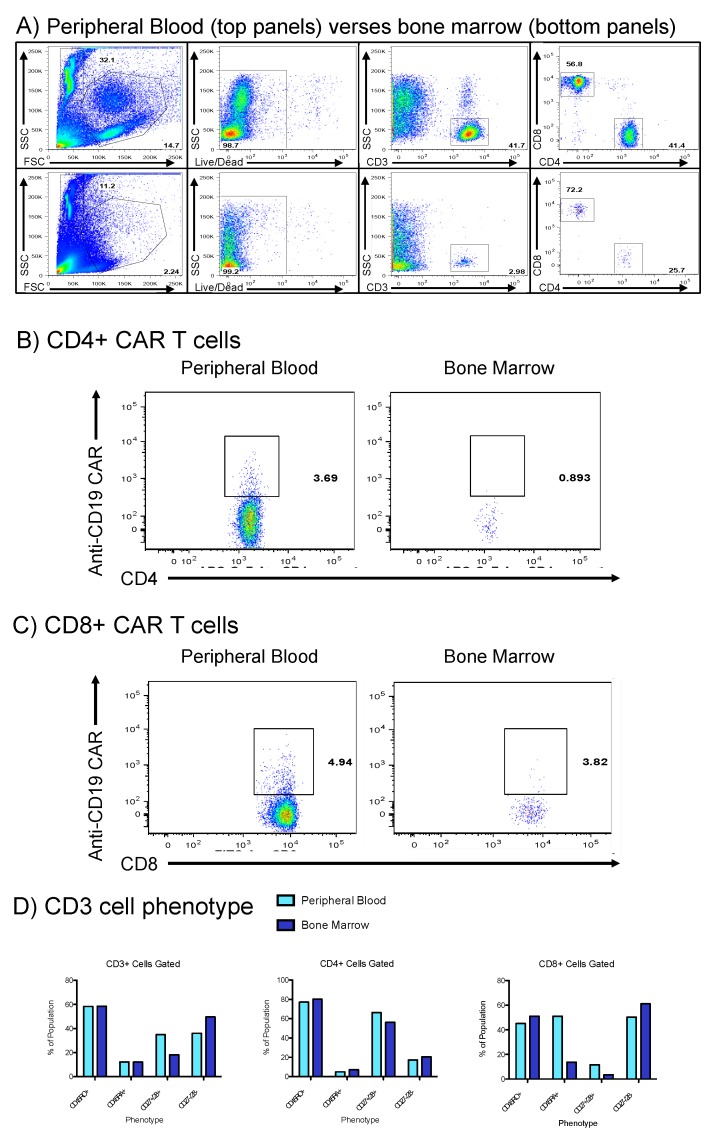
Comparison of bone marrow aspirates and peripheral blood during episode of pancytopenic aplasia. (**A**): Whole peripheral blood (top) was compared to bone marrow aspirates (bottom row) with qualitative finding of reduced marrow cellular content. (**B**,**C**): CD4+/CD8+ CAR T cells, respectably, were detectable in the peripheral blood and to a minimal extent in the marrow. (**D**): Comparison of memory/activated T cell (CD45RO+), naïve T cell (CD45RA+), and CD27/28 co-expression in bone marrow (dark blue) and peripheral blood (light blue).

**Figure 5 ijms-19-04118-f005:**
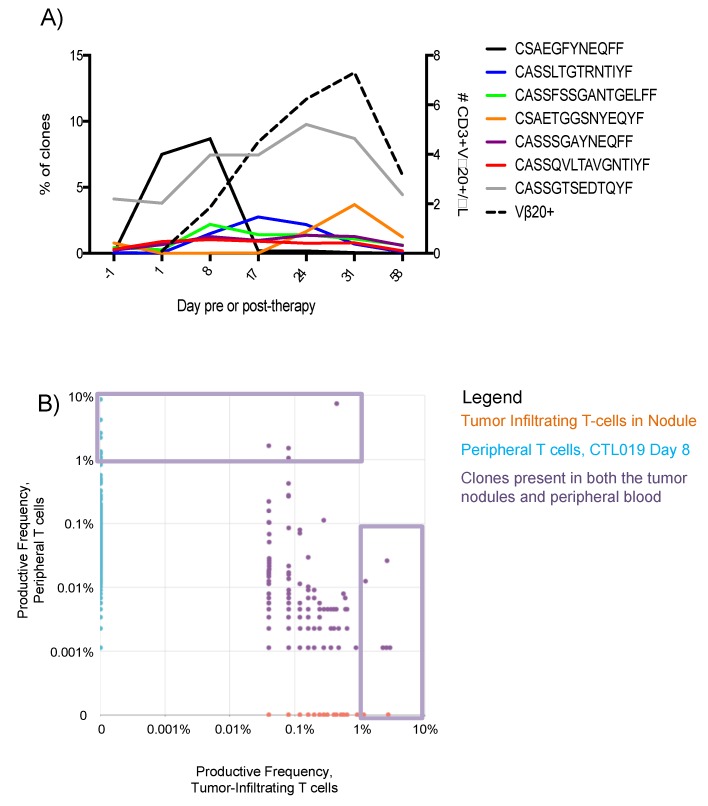
Transient expansion of T cell clones, present at high frequency within the tumor. (**A**): Deep sequencing of the TCRβ CDR3 regions as performed by Adaptive Biotechnologies (Seattle, WA, USA) on PBMCs collected one day prior to CAR T administration, then on post-infusion days 1, 8, 17, 24, 31, and 58. Deep sequencing data are presented as the productive frequency of clones containing the indicated amino acid sequences in the CDR3 region at each time point. The expansion of CAR T cells was determined by flow cytometry against antibody targeting the Vβ-20+ sequence in the CAR (dashed line). (**B**): Pair-wise scatter plot showing the productive frequency of the sum of frequencies of clones of T cells, based upon the CDR3 sequencing data. X-axis represents T cells collected from a scapular nodule on post-CAR day 10, while y-axis denotes peripheral blood collected on post-CAR day 8. The violet boxes denote preferential expansion of small numbers of T cell clones, denoted oligoclonal expansion.

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
