# Peer review of "Oligoclonal T Cells Transiently Expand and Express Tim-3 and PD-1 Following Anti-CD19 CAR T Cell Therapy: A Case Report"

_ijms, 2018, doi:10.3390/ijms19124118_

Round 1
Reviewer 1 Report
This very interesting case and study, well written. The manuscript show data from clinical case when patient died because of remission. The authors discovered that TIM-3 and PD-1 were highly expressed causing CAR-T/T cell exhaustion. Such cases are very important to report to improve future CAR-T therapies and increase patient survival.
Author Response
Point 1: This very interesting case and study, well written. The manuscript show data from clinical case when patient died because of remission. The authors discovered that TIM-3 and PD-1 were highly expressed causing CAR-T/T cell exhaustion. Such cases are very important to report to improve future CAR-T therapies and increase patient survival.
Response 1: Thank you for your interest in the case report. I appreciate your consideration of it for publication. Hopefully, this patient's case will inform hypothesis formation for future strategies to improve outcomes in CAR T cell therapy.
Reviewer 2 Report
Well written and presented case report of a patient with DLBCL treated with anti-CD19 CART-cells. The topic is extremelly interesting and the immune mechanisms of resistance to treatment need further clarification.
As authors mentioned, the major limitation is that this is a report of only 1 patieent and therefore it is almost imposible to make any conclusion regarding the mechanism of resistance.
For further improvement i suggest:
If authors collected data from responders then it will be very useful to present these data as well. In more detail:
1) Authors mention that CART cells represent only 2% of the total CD3 population in the specific patient while it usually represents 40% of total CD3 in patients with durable responses. Therefore i suggest to present data regarding expression of immune checkpoint receptors in responders
2) What was the percentage of immune senescent T-cells in responders?
Author Response
Point 1: Authors mention that CART cells represent only 2% of the total CD3 population in the specific patient while it usually represents 40% of total CD3 in patients with durable responses. Therefore I suggest to present data regarding expression of immune checkpoint receptors in responders.
Response 1: This is an excellent suggestion, and we have data which speak to this concern. Since this patient was originally enrolled in the clinical trial, JULIET, comparison of results from this single patient to responders in JULIET was possible. In JULIET, prior to CAR T manufacture, DLBCL tissue biopsy was performed on all patients at baseline, and this sample was subjected to quantitative immunofluorescence and AQUA analysis. Comparison of median baseline immune checkpoint levels to the expression of immune checkpoint molecules in this single patient is now discussed in paragraph 5 of the discussion section.
Point 2: What was the percentage of immune senescent T-cells in responders?
Response 2: This also is a superb question, but, unfortunately, the only data we have that directly answer the question comes from a previously published study from Dr. Waller's lab which showed that CD27-/CD28- T cells fail to expand in an ex vivo culture system. The parent clinical trial, JULIET, did not analyze this aspect of the patient T cell phenotype, and we did not collect patient sample for research analysis from any patients other than this single patient who experienced only a partial response. Therefore, to address this concern, I added discussion of a recent study which positively associate the frequency of CD27+/ CD45RO-/ CD8+ T cells prior to CAR T manufacture with remissions to paragraph 4. This question from the reviewer is certainly worth further exploration in future clinical trials and studies. I can certainly add more discussion if the reviewer feels this concern is inadequately addressed.
Round 2
Reviewer 2 Report
Authors replied adequately to my comments